# Enzyme-activating B-cell receptors boost antigen presentation to pathogenic T cells in gluten-sensitive autoimmunity

Rasmus Iversen [1,2,4] ✉, Julie Elisabeth Heggelund [1,2,4], Saykat Das[1,2], Lene S. Høydahl [1,2,3] & Ludvig M. Sollid [1,2] ✉

Autoantibodies against the enzyme transglutaminase 3 (TG3) are characteristic to the gluten-sensitive skin disorder dermatitis herpetiformis (DH), which is an extraintestinal manifestation of celiac disease. We here demonstrate that TG3-specific B cells can activate gluten-specific CD4[+] T cells through B-cell receptor (BCR)-mediated internalization of TG3-gluten enzyme-substrate complexes. Stereotypic anti-TG3 antibodies using *IGHV2-5/IGKV4-1* gene segments enhance the catalytic activity of TG3, and this effect translates into increased gluten presentation to T cells when such antibodies are expressed as BCRs. The crystal structure of TG3 bound to an *IGHV2-5/IGKV4-1* Fab shows that antibody binding to a β-sheet in the catalytic core domain causes the enzyme to adopt the active conformation. This mechanism explains the production of stereotypic anti-TG3 autoantibodies in DH and highlights a role for TG3-specific B cells as antigen-presenting cells for gluten-specific T cells. Similar boosting effects of autoreactive BCRs could be relevant for other autoimmune diseases, including rheumatoid arthritis.

Self-reactive immunoglobulins are involved in autoimmune diseases in the form of soluble autoantibodies and as membrane-bound B-cell receptors (BCRs). While autoantibodies against intracellular antigens often do not have a clear pathogenic effect, the successful use of B-cell depletion therapies to treat different autoimmune diseases suggests that B cells have important antibody-independent functions[1]. A key role of self-reactive B cells is likely to act as antigen-presenting cells for pathogenic CD4[+] T cells, since BCR-mediated uptake allows efficient presentation of antigen-derived peptides on HLA class II molecules[2,3]. Interactions between antigen-specific T cells and B cells drive their mutual activation and lead to generation of effector T cells and antibody-producing plasma cells.

In the gluten-sensitive enteropathy celiac disease (CeD), production of autoantibodies against the enzyme transglutaminase (TG) 2 is explained by a hapten-carrier-like model, where TG2-specific B cells bind complexes of TG2 and gluten, allowing interaction with gluten-

specific CD4[+] T cells[4]. This mechanism selects for BCRs that favor formation of TG2-gluten enzyme-substrate complexes, thereby shaping the autoantibody response[5–7]. The blistering skin disorder dermatitis herpetiformis (DH) is an extraintestinal manifestation of CeD characterized by autoantibodies against TG3 in addition to TG2[8]. Both gluten-specific CD4[+] T cells[9,10] and TG3-specific gut plasma cells[11,12] have been demonstrated in DH patients, suggesting that the anti-TG3 response could be induced in the gut through similar mechanisms as described for the anti-TG2 response in CeD.

TGs are a family of $Ca^{2+}$ dependent enzymes that catalyze acyl transfer reactions, where Gln residues of polypeptides are modified by transamidation (resulting in crosslinking via isopeptide bond formation) or deamidation (resulting in Gln to Glu conversion) in a sequence-specific fashion[13,14]. Deamidation is critical for generation of antigenic gluten peptides, which are recognized by gluten-specific CD4[+] T cells when presented in complex with disease-associated HLA-DQ allotypes

[1]Norwegian Coeliac Disease Research Centre, Institute of Clinical Medicine, University of Oslo, Oslo, Norway. [2]Department of Immunology, Oslo University Hospital - Rikshospitalet, Oslo, Norway. [3]Present address: Nextera AS, Oslo, Norway. [4]These authors contributed equally: Rasmus Iversen, Julie Elisabeth Heggelund. ✉e-mail: rasmus.iversen@medisin.uio.no; l.m.sollid@medisin.uio.no

(in particular HLA-DQ2.5)[15]. All TGs comprise an N-terminal domain, a catalytic core domain and two C-terminal domains (C1 and C2)[13]. Uniquely among TGs, TG3 must be proteolytically cleaved at a linker between the catalytic core domain and the C1 domain to become catalytically active[16]. The crystal structure of TG3 with a substrate-mimicking inhibitor bound at the active site revealed that the C1 and C2 domains depart from the rest of the enzyme and that a β-sheet in the catalytic core domain shifts position upon binding of substrate[17]. TG3-specific monoclonal antibodies cloned from gut plasma cells of DH patients all recognize this enzyme-substrate intermediate as well as the zymogen form of TG3, and they exhibit no cross-reactivity with TG2[12]. The antibodies target conformational epitopes within three distinct regions of TG3, termed epitope 1-3. Epitope 3 antibodies are characterized by usage of *IGHV2-5* in combination with *IGKV4-1*. The same V genes were found to be enriched in TG3-specific serum IgA of DH patients, suggesting that this type of antibodies dominates the autoantibody response[12].

By recombinant expression of DH patient-derived BCRs, we here demonstrate that TG3-specific B cells of all three epitope groups can present deamidated gluten peptides to T cells through BCR-mediated uptake of TG3-gluten complexes. Moreover, we find that epitope 3 antibodies augment the enzymatic activity of TG3, translating into increased uptake and presentation of gluten peptides by B cells expressing epitope 3 BCRs. The crystal structure of TG3 in complex with an epitope 3 Fab (DH63-A02) unveils the mechanism underlying this phenomenon. By binding to the critical β-sheet within the catalytic core domain, the antibody triggers a conformational shift, inducing the active enzyme state. Our findings explain the autoantibody response in DH and highlights a role for B cells with enzyme-activating BCRs in activation of pathogenic gluten-specific T cells. Autoreactive B cells that modulate the function of their target antigen may play a similar role in other autoimmune diseases.

## Results

### TG3-mediated gluten deamidation facilitates interactions between TG3-specific B cells and gluten-specific T cells

In both CeD and DH, gluten-specific T cells recognize deamidated peptides that reflect the catalytic activity of TG2[10,18,19]. To test if TG3 is also able to form deamidated peptides that can activate disease-relevant T cells, we stimulated gluten-reactive CeD patient-derived T-cell lines with a whole-gluten digest that had been treated with active TG2 or TG3 (Fig. 1a and Supplementary Fig. 1). Both enzymes facilitated T-cell activation with similar efficiency, demonstrating that TG3 alike TG2 can generate relevant T-cell epitopes. To further compare TG2- and TG3-mediated deamidation, we took advantage of T-cell receptor (TCR)-transduced BW58 mouse hybridoma T cells specific to two immunodominant gluten epitopes. Due to the deamidation sensitivity of the TCRs, activation of these cells can be used to assess deamidation of specific Gln residues within the epitope sequences. Stimulation of the T cells with synthetic gluten peptides that had been treated with either TG2 or TG3 revealed that TG3 is less efficient in deamidating the DQ2.5-glia-α2 epitope of α-gliadin (Fig. 1b) but almost as efficient as TG2 in deamidating the DQ2.5-glia-ω2 epitope of ω-gliadin (Fig. 1c). We therefore decided to use DQ2.5-glia-ω2-specific T cells when setting up T cell-B cell collaboration assays.

To test if TG3-specific B cells can present peptides to gluten-specific T cells, we generated a BCR-transduced A20 mouse lymphoma B-cell line expressing an anti-TG3 (DH63-B02) IgD BCR in combination with HLA-DQ2.5 (Fig. 1d and Supplementary Fig. 2). The crystal structure of TG3 bound to the Fab of DH63-B02 has previously been solved, and the epitope was demonstrated to be localized in the catalytic core domain away from the active site[17]. We therefore hypothesized that BCR binding to TG3 would not interfere with the enzyme's ability to form complexes with gluten peptides and that it would thus facilitate peptide uptake by the B cells. Comparison of the TG3-specific cells with similar cell lines expressing anti-TG2 (679-14-E06) or control (693-2-F02) BCR revealed that only cells specific to the relevant TG induce T-cell activation when pulsed with gluten peptide in combination with active TG2 or TG3 (Fig. 1e, f). These results confirm that both TG2- and TG3-specific B cells can present deamidated gluten peptides to T cells through BCR-mediated uptake of cognate TG enzyme in complex with gluten substrate.

### Anti-TG3 epitope 3 antibodies enhance TG3-mediated deamidation

Knowing that the enzymatic activity of TG3 can facilitate interactions between TG3-specific B cells and gluten-specific T cells, we next addressed if targeting of individual TG3 epitopes affects TG3-mediated deamidation. For this purpose, we took advantage of a panel of TG3-specific monoclonal antibodies that has previously been generated from gut plasma cells of DH patients. The antibodies comprise ten different clonotypes that were assigned to three individual epitope groups (epitope 1-3) based on competition for TG3 binding[12]. By using T-cell activation as readout of peptide deamidation, we assessed if the antibodies influence enzyme activity (Fig. 2a). None of the ten antibodies inhibited TG3-mediated deamidation, consistent with a model where preserved catalytic activity of BCR-bound TG3 is necessary for TG3-specific B cells to interact with gluten-specific T cells. Interestingly, two of the antibodies caused a dose-dependent increase in deamidation when incubated with active TG3 (Fig. 2a, b). Both antibodies belong to epitope group 3 and use the canonical *IGHV2-5/IGKV4-1* gene segments. To explore the potential effect of the antibodies on the conformational state of TG3, we incubated proteolytically cleaved TG3 with antibodies belonging to each of the three epitope groups and performed pull-down of the resulting complexes (Fig. 2c). The two proteolytic fragments of TG3 remained attached to each other via non-covalent interactions when incubated with epitope 1 or epitope 2 antibodies. However, the two epitope 3 antibodies only pulled down the fragment consisting of the N-terminal domain and the catalytic core domain, indicating that binding to epitope 3 leads to dissociation of the C1 and C2 domains from the rest of the enzyme. Since we have previously observed that substrate-bound TG3 is devoid of the C1C2 fragment[17], these results suggest that epitope 3 antibodies augment TG3 activity by inducing the catalytically active enzyme conformation.

### Epitope 3 Fab induces the active conformation of TG3 by binding to a critical β-sheet in the catalytic core domain

In the absence of substrate, cleaved TG3 with bound $Ca^{2+}$ assumes a conformation, where the two C-terminal domains remain attached to the catalytic core domain through non-covalent interactions, thereby occluding the active site (Fig. 3a)[17,20]. Upon binding of substrate, a β-sheet in the catalytic core domain shifts position, and the C1C2 fragment dissociates (Fig. 3b)[17]. To understand the effect of epitope 3 antibodies on TG3 activity, we solved the co-crystal structure of an epitope 3 Fab (DH63-A02) bound to cleaved TG3 with or without a substrate-mimicking inhibitor attached to Cys273 in the active site (Fig. 3c, d, and Supplementary Fig. 3 and Supplementary Table 1). Except for the presence or absence of the peptide inhibitor, the two structures are virtually identical (RMSD of Cα atoms 0.52 Å), and the C-terminal domains are lacking from both. The structures reveal that DH63-A02 binds to the critical catalytic core β-sheet and that its epitope does not overlap with the previously characterized epitope of DH63-B02 (epitope 2)(Supplementary Figs. 4a–c). With the exception of Leu100 in CDR-L3, all heavy and light chain residues that are involved in electrostatic interactions with TG3 are germline encoded (Fig. 3e and Supplementary Table 2). The identified Fab-TG3 interactions thus explain the selective usage of *IGHV2-5/IGKV4-1* among anti-TG3 epitope 3 antibodies.

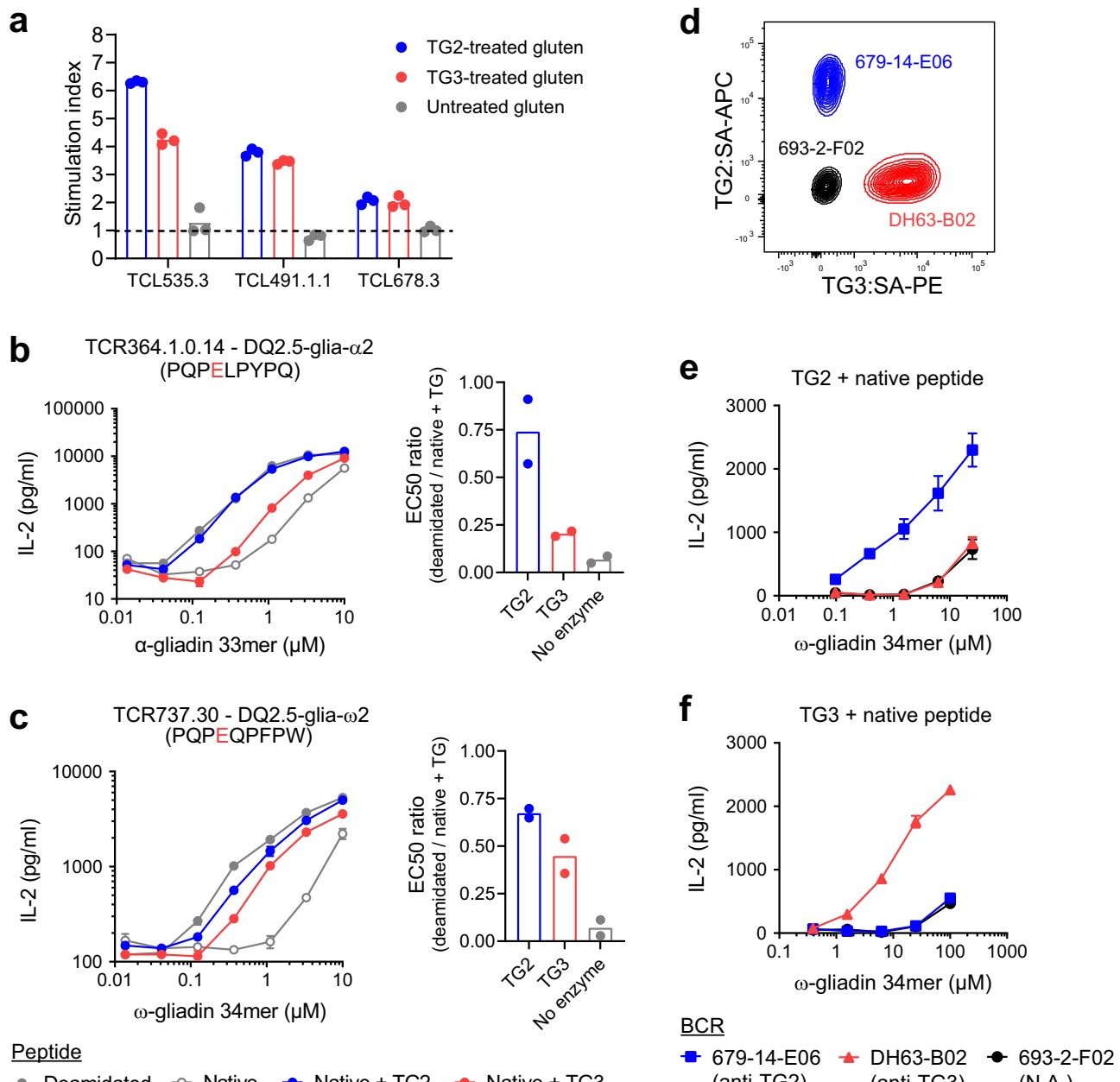

**Fig. 1 | TG3-specific B cells can present deamidated gluten epitopes to T cells.**
**a** Stimulation of three human gluten-reactive T-cell lines with non-deamidated gluten or gluten pre-treated with active TG2 or TG3 assessed by ³H-thymidine incorporation. The stimulation index was calculated as the signal obtained with gluten divided by the signal obtained with medium only. Bar heights indicate means of culture triplicates from the same experiment. **b**, **c** Stimulation of TCR-transduced BW58 T cells specific for the immunodominant gluten epitopes DQ2.5-glia-α2 (**b**) or DQ2.5-glia-ω2 (**c**) by HLA-DQ2.5-expressing A20 B cells assessed by release of murine IL-2. The cells were co-cultured in the presence of gluten peptides harboring the deamidated or non-deamidated (native) epitope sequences (shown in brackets with deamidation sites indicated in red). The native peptides were either added directly or pre-treated with active TG2 or TG3. Dose-response curves (left

panels) were used for calculation of EC50 values (right panels). Symbols represent means of culture duplicates, and error bars indicate range. Bar heights indicate means of two experiments. **d** Flow cytometry plot showing staining of three BCR-transduced A20 B-cell lines with recombinant biotinylated TG2 and TG3 coupled to fluorescently labeled streptavidin (SA). The BCRs were generated from gut plasma cells that were specific for TG2 (679-14-E06) or TG3 (DH63-B02) or had unknown specificity (693-2-F02). **e**, **f** Stimulation of DQ2.5-glia-ω2-specific T cells by A20 B cells expressing HLA-DQ2.5 and BCRs with the indicated specificities. N.A. not applicable. The A20 cells were pulsed with native gluten peptide in combination with active TG2 (**e**) or TG3 (**f**) prior to co-culture with T cells. Symbols represent means of culture duplicates, and error bars indicate range. Results from one of two experiments are shown. Source data are provided as a Source Data file.

The structures of DH63-A02 bound to cleaved TG3 predict a clash between DH63-A02 and the C1 domain if the C1C2 fragment had still been present (Supplementary Figs. 4d, e). Binding of antibody to epitope 3 is therefore expected to induce conformational changes also in zymogen TG3. In support of this notion, we observed that overnight incubation of zymogen TG3 with two epitope 3 Fabs resulted in formation of high-MW covalent complexes, indicating an extent of TG3-

mediated protein crosslinking (transamidation) (Supplementary Fig. 5). Similar complexes were not observed with epitope 1 or epitope 2 Fab, suggesting that binding to epitope 3 is required for zymogen TG3 to display low-level transamidation activity. This effect is likely explained by a Fab-induced move of the C-terminal domains to uncover the active site, since the catalytic core β-sheet and the C1 domain will be forced apart, when an antibody binds to epitope 3.

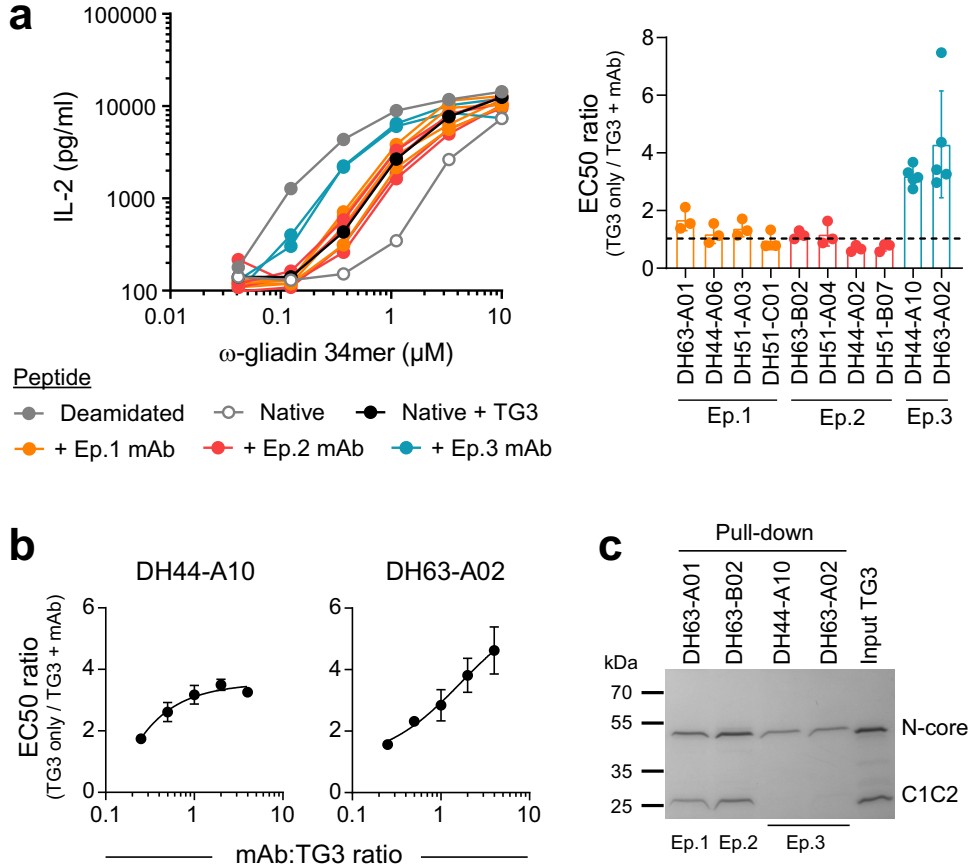

**Fig. 2 | Effect of anti-TG3 antibodies on TG3 activity. a** Stimulation of DQ2.5-glia-ω2-specific BW58 T cells by HLA-DQ2.5-expressing A20 B cells in the presence of deamidated or native gluten peptide. The native peptide was either added directly or pre-incubated with active TG3 with or without bound anti-TG3 monoclonal antibodies (mAbs, $n = 10$) targeting three different epitopes (Ep. 1-3). Dose-response curves (left panel) were used for calculation of EC50 values (right panel). Bar heights indicate means, and error bars indicate SD based on 3 (Ep.1 and Ep.2) or 5 (Ep.3) experiments. **b** Titratable effect of two epitope 3 mAbs on TG3-mediated deamidation assessed by activation of gluten-specific T cells as in (**a**). Symbols represent means of two experiments, and error bars indicate range. **c** SDS-PAGE analysis of proteolytically cleaved TG3 before and after pull-down with anti-TG3 mAbs targeting different epitopes. One of five experiments is shown. Source data are provided as a Source Data file.

Collectively, the structures of TG3 with bound DH63-A02 demonstrate that the antibody autonomously triggers the conformational changes associated with TG3 activation, thus explaining why epitope 3 antibodies enhance the catalytic activity of the enzyme.

## B cells specific to epitope 3 are superior in presentation of gluten peptide to T cells

To explore how targeting of individual TG3 epitopes affects B-cell activation in DH, we constructed BCR-transduced A20 cell lines representing each of the three epitope groups (Fig. 4a). One of the epitope 3-targeting cell lines (DH63-A02) bound lower amounts of TG3 than the other cells. This behavior is ascribed to a combination of lower overall BCR expression (Fig. 4a) and slightly lower affinity of the epitope 3 BCRs compared to the BCRs targeting epitope 1 or epitope 2[17]. When the cells were incubated with active TG3, we observed that the IgD BCRs were incorporated into high-MW covalent complexes as a result of TG3-mediated transamidation (Fig. 4b). Interestingly, only the fully glycosylated form of the δ heavy chain containing O-linked glycans in the hinge was crosslinked by TG3 (Figs. 4b and Supplementary Fig. 6)[21,22]. As judged by depletion of monomeric δ chains, this crosslinking was more efficient with epitope 3 BCRs than with epitope 1 or epitope 2 BCRs.

Since bridging of surface BCR molecules is known to facilitate signal transduction in B cells, we next assessed the effect of TG3 on BCR signaling by measuring release of intracellular $Ca^{2+}$ (Fig. 4c–e). For the two epitope 3 BCRs, stimulation with activated TG3 led to elevated $Ca^{2+}$ release (Fig. 4d), and the peak of the signal was delayed compared to stimulation with zymogen TG3 (Fig. 4e). These observations are likely explained by TG3-mediated transamidation leading to covalent BCR crosslinking and enhanced, but slower, signaling as compared to simple antigen binding. Since binding to epitope 3 augments the catalytic activity of TG3, this effect on BCR signaling was more pronounced for B cells targeting epitope 3.

BCR-mediated modulation of TG3 activity potentially affects B-cell uptake of TG3-gluten complexes. To test this concept, we loaded the B cells with activated or zymogen TG3 followed by incubation with fluorescently labeled gluten peptide and assessed peptide uptake by flow cytometry (Fig. 4f). Despite binding lower amounts of TG3 (Fig. 4a), the epitope 3 B cells acquired more gluten peptide than epitope 1 or epitope 2 B cells when active TG3 was bound to the BCR (Fig. 4f, g). These results indicate that TG3 more readily forms enzyme-substrate complexes with gluten when bound to an epitope 3 BCR. Even when loaded with zymogen TG3, the two epitope 3 B cells showed uptake of low amounts of gluten peptide (Fig. 4f), consistent with our observation that epitope 3 antibodies cause the uncleaved enzyme to acquire a degree of catalytic activity (Supplementary Fig. 5).

To address whether targeting of individual TG3 epitopes also affects antigen presentation to T cells, we pulsed the BCR-transduced A20 cells with TG3 in combination with gluten peptide and assessed antigen presentation by stimulation of gluten-specific BW58 T cells

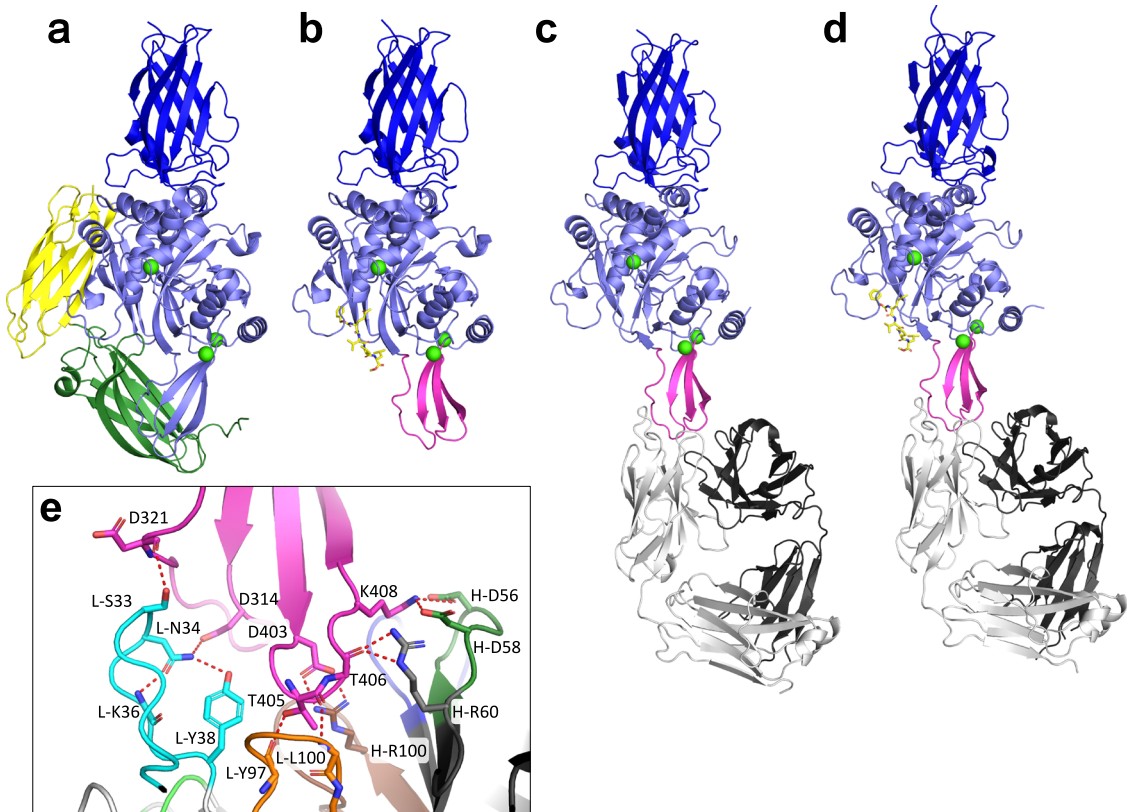

**Fig. 3 | Fab DH63-A02 induces the catalytically active conformation of TG3.**
**a**, **b** Previously solved crystal structures of proteolytically cleaved TG3 without bound substrate (**a**, PDB ID: 8OXW) or with a substrate-mimicking irreversible inhibitor (Z-DON, yellow stick) attached to the active site Cys (**b**, PDB ID: 8OXX). The structures were solved with bound anti-TG3 Fab DH63-B02, but the Fab is omitted in these representations. Color coding: N-terminal domain, dark blue; catalytic core domain, light blue; C1 domain, green; C2 domain, yellow; and calcium ions, green spheres. A β-sheet that is repositioned upon binding of substrate and departure of C1C2 is highlighted in magenta in (**b**). **c**, **d** Crystal structures of cleaved TG3 in complex with anti-TG3 Fab DH63-A02 (heavy chain, dark gray; and light chain, light gray). The structures were solved without inhibitor (**c**, PDB ID: 8RMX) or with Z-DON bound in the active site (**d**, PDB ID: 8RMY). **e** Closeup of paratope-epitope interactions between Fab DH63-A02 and TG3 with bound Z-DON (PDB ID: 8RMY). Direct H-bonds are depicted with red broken lines. Color coding for Fab CDR loops: CDR-H1, dark blue; CDR-L1, cyan; CDR-H2, dark green; CDR-L2, light green; CDR-H3, brown; and CDR-L3, orange.

(Fig. 4h). As expected, all TG3-specific B cells presented gluten to the T cells when incubated with active TG3. Importantly, T-cell stimulation was more efficient if the B cells expressed an epitope 3 BCR, in agreement with enhanced complex formation between BCR-bound TG3 and gluten substrate. Taken together, our results suggest that TG3-specific B cells targeting epitope 3 are preferentially activated in DH, due to the augmented catalytic activity of BCR-bound TG3. The enzyme activity plays an important role in B-cell activation, as it both facilitates stimulation via BCR crosslinking and allows gluten uptake through formation of TG3-gluten enzyme-substrate complexes. As a result, epitope 3-targeting B cells will present more deamidated peptide and receive more help from gluten-specific T cells than other TG3-specific B cells.

## Discussion

By taking advantage of monoclonal antibodies generated from gut plasma cells of DH patients, we here demonstrate that BCR-bound TG3 can serve as a sink for gluten peptides and thereby facilitates crosstalk between TG3-specific B cells and gluten-specific CD4⁺ T cells. A similar hapten-carrier-like mechanism as described for anti-TG2 autoantibodies in CeD thus explains the gluten-dependent anti-TG3 response in DH. Another important aspect of our study is that autoantibodies directed against a particular TG3 epitope (epitope 3) can induce the active enzyme conformation. By means of X-ray crystallography, the structural basis for this induced activity was resolved. We show that the prototypic epitope 3 antibody DH63-A02

binds to a β-sheet in the catalytic core domain of TG3 and thereby induces a conformational change, which leads to exposure of the active site. This activity-inducing effect explains both why TG3 exerts higher deamidating activity when epitope 3 antibodies are present in solution and why epitope 3-targeting B cells can present more gluten peptide to T cells than other TG3-specific B cells. Enhanced antigen presentation to T cells likely drives preferential activation of epitope 3-targeting B cells in DH. Since epitope 3 antibodies are characterized by usage of *IGHV2-5/IGKV4-1* gene segments, our findings give the molecular underpinning for the previously observed preference for these V genes among TG3-specific autoantibodies[12]. While it would be ideal to demonstrate a pathogenic role of epitope 3-targeting antibodies/BCRs by assessing their potential association with more rapidly progressing or more aggressive disease, at this point unfortunately there exists no clinical tool to grade severity of DH.

Similar to the TG3-activating antibodies described here, TG2-specific antibodies in CeD differentially affect TG2 activity and, as a result, the efficiency of gluten peptide presentation by TG2-specific B cells[5,6]. However, the structural basis for antibody-mediated effects on TG2 activity is yet to be solved. The observed mechanism of autoantibody formation, where antibodies/BCRs enhance the activity of disease-implicated enzymes likely goes beyond DH and CeD. Rheumatoid arthritis (RA) is a disorder with particularly striking parallels. RA patients have disease-specific antibodies to citrullinated peptides as well as to the enzymes that mediate the citrullination of Arg

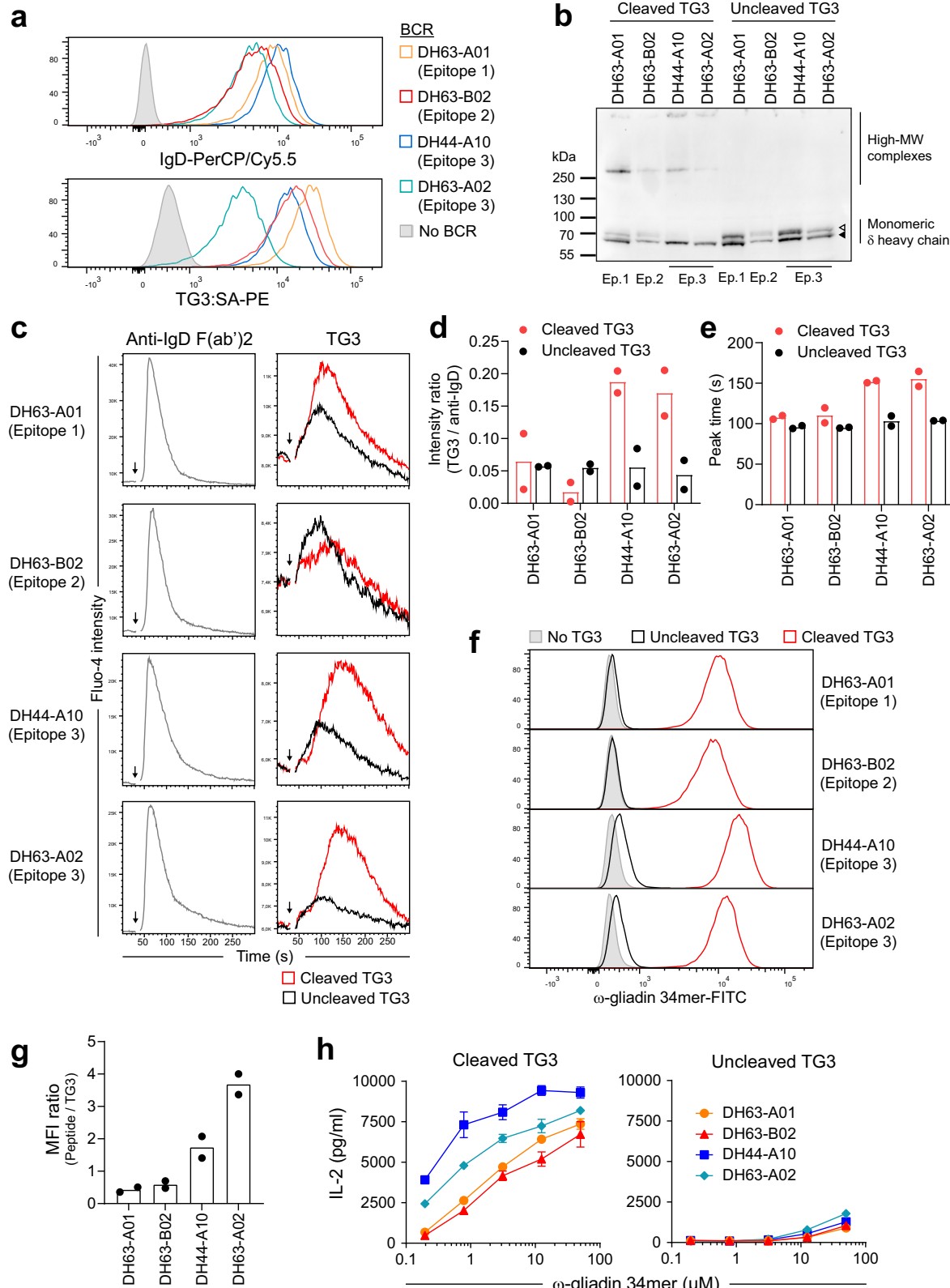

residues, peptidylarginine deiminase (PAD) 2 and PAD4[23,24]. Parallel to the existence of antibodies that increase the catalytic activity of TG2 in CeD and TG3 in DH, PAD-reactive antibodies that increase the activity of PAD4 have been described[25]. Presence of such antibodies was shown to be associated with erosive arthritis, i.e., more severe disease, suggesting that the antibodies might be particularly important in the pathogenesis. Given the findings in CeD and DH, it is tempting to

suggest that enzyme-activating BCRs in RA might boost interactions between PAD4-specific B cells and citrulline-specific CD4+ T cells.

Although both TG2 and TG3 can deamidate gluten peptides that are recognized by disease-relevant T cells, their fine specificity differs somewhat[26,27]. A difference in substrate specificity is supported by distinctions in the active site structures between TG2 and TG3[17]. Here, we demonstrate that the two enzymes deamidate the DQ2.5-glia-ω2

**Fig. 4 | Effects of TG3 binding by TG3-specific BCRs. a** Flow cytometry histograms showing staining of BCR-transduced A20 B cell lines targeting three different TG3 epitopes (Ep.1-3). The cells were assessed for surface IgD expression or binding to monomeric TG3 by staining with recombinant biotinylated TG3 followed by PE-conjugated streptavidin (SA-PE). **b** Western blot showing detection of IgD in A20 cell lysates. The cells were incubated with active (cleaved) or zymogen (uncleaved) TG3 prior to lysis and reducing SDS-PAGE. Two monomeric δ heavy chain bands are visible, representing the fully glycosylated form with both *O*-linked and *N*-linked glycans (open triangle) and a partially glycosylated form with *N*-linked glycans only (closed triangle). The upper band disappears or is diminished after incubation with cleaved TG3 due to incorporation into high-MW complexes, some of which are too large to enter the gel. One of three experiments is shown. **c-e**, Calcium flux assay showing representative flow cytometry signals (**c**) and quantification of peak heights (**d**) and peak time points (**e**). After recording baseline signal for 30 s, BCR-transduced A20 cells were stimulated with cleaved or uncleaved TG3 (indicated by arrows). A crosslinking anti-IgD F(ab')2 was included as positive control for BCR-induced $Ca^{2+}$ release. Bar heights indicate means of two experiments. **f, g** Uptake of fluorescently labeled gluten peptide by BCR-transduced A20 cells. Representative histograms of A20 cells pre-incubated in the presence or absence of cleaved or uncleaved TG3 are shown in (**f**). Peptide uptake relative to the amount of BCR-bound TG3 (**g**) was assessed as the ratio between median fluorescence intensity (MFI) of cell-associated peptide in presence of cleaved TG3 (**f**) and MFI of BCR-bound TG3 (**a**). Bar heights indicate means of two experiments. **h**, Stimulation of DQ2.5-glia-ω2-specific BW58 T cells by BCR-transduced A20 B cells expressing HLA-DQ2.5. The cells were pulsed with native gluten peptide in combination with cleaved or uncleaved TG3 prior to co-culture with T cells. Symbols represent means of culture triplicates, and error bars indicate SD. Results from one of two experiments are shown. Source data are provided as a Source Data file.

epitope of ω-gliadin almost equally well, but TG2 is much more efficient in deamidating the DQ2.5-glia-α2 epitope of α-gliadin. Conversely, there may exist gluten peptides that are substrates of TG3 but not TG2 and that give rise to T-cell epitopes, which are unique to DH patients. Potentially, such epitopes can be identified by using TG3-specific B cells as antigen-presenting cells for T cells of DH patients.

Antibodies specific for well-defined epitopes tend to use selected heavy and light chain gene segments in a stereotyped manner[28]. While this selection bias can be explained by the fact that some immunoglobulin genes possess an inherent germline-encoded ability to bind antigenic structures[29], our current findings and previous studies of CeD[5,7] demonstrate that binding of antigen to the BCR is not the only determining factor for B-cell activation and antibody production. The B cell must also be able to productively interact with T cells. TG-gluten complexes capable of driving B-cell interactions with gluten-specific T cells can in principle take two forms: an enzyme-substrate intermediate or an isopeptide-linked transamidation product[30]. The former complex is inevitably formed by both TG2 and TG3 when there is substrate turnover. The isopeptide-linked transamidation product, on the other hand, could only be detected for TG2 and not TG3 under conditions of relatively high concentrations of recombinant enzyme[26]. Importantly, in lysates of intestinal epithelial cells, isopeptide-linked TG2-gluten complexes could not be detected, even though the endogenous enzyme was still able to drive collaboration between TG2-specific B cells and gluten-specific T cells[31]. Thus, under physiological conditions characterized by low concentration of active enzyme and high levels of competing protein substrate, the most relevant TG-gluten complex appears to be the enzyme-substrate intermediate for both the anti-TG2 and the anti-TG3 response. Importantly, upon BCR-mediated uptake into endosomes, this intermediate will be converted into free enzyme and free deamidated peptide. Hence, this mechanism directly couples B-cell uptake of antigen to presentation of deamidated peptide, and it implies that BCR binding does not block TG catalytic activity. It follows that the most efficient presentation is achieved when BCRs can augment formation of TG-gluten enzyme-substrate complexes[5,6].

Our studies of CeD and DH point to autoreactive B cells as key antigen presenting cells for pathogenic CD4+ T cells. The findings that anti-TG3 serum antibodies primarily belong to the IgA class and that TG3-specific plasma cells are present in the gut lamina propria of DH patients suggest that the response has a mucosal origin and that T cell-B cell interactions take place in gut-associated lymphoid tissues[8,11,12]. Whether gluten-reactive CD4+ T cells are also involved in formation of skin lesions in DH needs to be established. A study on circulating gluten-reactive T cells in five DH patients reported a single case where a substantial fraction of the cells had a skin-homing phenotype[10]. Since dietary gluten peptides are systemically distributed[32], and antigenic TG3 is present in the skin[8], it is possible that T cell-B cell interactions can also take place in this compartment.

Regardless of where T cell-B cell collaboration anatomically takes place, it remains a key observation that interacting T cells and B cells do not need to recognize the same antigen molecule. In CeD and DH, complex formation between self- and non-self-antigens facilitates presentation of foreign antigen by autoreactive B cells. Similar processes may also occur in other autoimmune diseases.

## Methods

### Recombinant TGs

Recombinant human TG2 and TG3 were produced in Sf+ insect cells with an N-terminal His6-tag followed by an Avi-tag for site-specific biotinylation. The proteins were biotinylated by co-expression of BirA biotin-protein ligase and purified from cell lysates by Ni-NTA affinity chromatography as previously described[12]. The enzymes were dialyzed against 20 mM Tris-HCl, pH 7.4, 300 mM NaCl, 1 mM DTT, 1 mM EDTA and supplemented with 10% (v/v) glycerol, before they were stored at −80 °C. To obtain the active form of TG3, the zymogen was cleaved with dispase I (Sigma) or cathepsin L (R&D Systems). The two proteases both cleave the linker between the catalytic core domain and the C1 domain of TG3, and they give almost identical proteolytic fragments[33]. Dispase I digestion was conducted in Tris-buffered saline (TBS) supplemented with 5 mM $CaCl_2$ by adding 0.1 mg dispase I per mg of TG3 followed by incubation for 30 min at 37 °C. Cathepsin L digestion was conducted in 100 mM MES, pH 6 supplemented with 10 µg/ml cathepsin L. After incubation for 30 min at 37 °C, cathepsin L was inactivated by addition of E-64 protease inhibitor (Sigma) to a concentration of 10 µM.

### Recombinant antibodies and Fabs

Anti-TG3 antibodies and Fabs were expressed as human IgG1 molecules in Expi293-F cells as previously described[12,17]. To obtain Fabs from full length antibodies, the heavy chain variable regions were subcloned into a constant region vector containing a stop codon after Asp6 (IMGT numbering) of the IgG1 hinge[6]. Full length antibodies were purified from culture supernatants on a protein G column (Cytiva), whereas kappa light chain Fabs were purified on a protein L column (Cytiva), and lambda light chain Fabs were purified on CaptureSelect LC-lambda (Hu) affinity matrix (Thermo). The proteins were eluted with 0.1 M Glycine-HCl, pH 2.5 and immediately pH neutralized. The purified proteins were concentrated and buffer exchanged into TBS (antibodies) or PBS (Fabs) using Vivaspin 20 centrifugal concentrators with 30 kDa (antibodies) or 10 kDa (Fabs) cut-offs (Sartorius).

### Chromatographic analysis of zymogen TG3 + anti-TG3 Fabs

Zymogen TG3 was incubated overnight with anti-TG3 Fabs in size-exclusion chromatography (SEC) running buffer (20 mM Tris-HCl, pH 8, 300 mM NaCl, 5 mM $CaCl_2$) at 4 °C, using 0.3 mg TG3 and a 1:1.1 molar ratio of TG3:Fab. The samples were analyzed by SEC, using a Superdex 200 10/300 column (GE Healthcare) at a flow rate of 0.75 ml/min in

running buffer. Fractions of 0.4 ml were collected and analyzed by reducing or non-reducing SDS-PAGE.

## Preparation of complexes for crystallization

Zymogen TG3 was digested with dispase I followed by separation of the cleaved enzyme from dispase I by SEC as described above, using a flow rate of 0.5 ml/min. Cleaved TG3 was mixed with Fab DH63-A02 at a 1:1.1 molar ratio and incubated overnight at 4 °C. The TG3 + Fab complex was purified by SEC, and the resulting peak fractions were pooled and concentrated to 8.9 mg/ml using a Vivaspin 500 centrifugal concentrator with a 10 kDa cut-off (Sartorius). To generate complexes of inhibitor-bound TG3 and Fab, the irreversible active-site inhibitor "Z-DON" (Z-DON-Val-Pro-Leu-OMe, Zedira) was included in the incubation step with Fab DH63-A02 at 25:1 molar ratio of Z-DON:TG3. The reaction mixture was incubated overnight at 23 °C followed by SEC purification of the resulting TG3 + Fab + Z-DON complex. Collected peak fractions were pooled and concentrated to 21.9 mg/ml.

## Crystallization of TG3-Fab complexes

The initial crystallization hits for the TG3 + Fab + Z-DON complex were obtained in the JCSG+ screen (Molecular Dimensions) well B6, using a sitting-drop setup dispensed by a Mosquito robot (SPT Labtech). These initial small crystals were used to create microseeds by crushing the crystals with a seed bead (Hampton Research) according to the manufacturer's instructions. The microseeds were used in subsequent hanging-drop optimization experiments inspired by random microseed matrix screening (rMMS)[34]. Both complexes were crystallized in 0.15 M phosphate-citrate, pH 5.0, 40% (v/v) ethanol, 5% (v/v) PEG 400. The TG3 + Fab complex was cross-seeded with microseeds of TG3 + Fab + Z-DON. Diffraction-quality crystals were obtained within 1-5 days. For both setups, the crystals were cryo-protected with 20% (v/v) ethylene glycol and flash-cooled in liquid nitrogen.

## Data collection and refinement

Diffraction data were collected at the ESRF beam line ID23-1 (24.01.2023) and the MAX IV beam line BioMAX (16.02.2023). The experimental session from ID23-1 can be traced through the DOI 10.15151/ESRF-ES-1022934233. The data collection and refinement statistics are shown in Supplementary Table 1. The data from ESRF were indexed, scaled and merged by the ESRF autoprocessing software, before the unmerged and unscaled files were reprocessed using AIMLESS[35] in the CCP4 software suite[36]. The data set from MAX IV was processed using DIALS[37] in the CCP4 software suite and subsequently processed by AIMLESS. Both complexes crystallized in space group I2. The structures could also be solved in the corresponding space group C2, but I2 was chosen since it gave the smallest β-angle. The structures were solved using molecular replacement by the program Phaser MR[38]. The TG3 + Fab + Z-DON structure was solved using the TG3 structure from PDB ID: 8OXX[17] and an Alphafold Colabfold[39] model of Fab DH63-A02 as molecular replacement search models. There are two copies of the complex in the asymmetric unit. Parts of the Alphafold Fab model was not optimally placed in the structure and had to be manually rebuilt in Coot[40]. The TG3 + Fab structure was solved using the refined TG3 + Fab + Z-DON structure as search model.

The structures were refined using REFMAC5[41] and manually optimized using Coot. Strong positive difference electron density was observed in the three $Ca^{2+}$ binding sites identified in previous TG3 structures, and in the active site for the TG3 + Fab + Z-DON structure. The Z-DON molecule and restraints library file were prepared using an isomeric SMILES string in AceDRG[42]. Both datasets were refined using local non-crystallographic symmetry (NCS) and translation/libration/screw-motion (TLS) restraints in REFMAC5. In the final stages, the structures were refined using ProSMART[43] restraints for TG3 from PDB ID: 8OXW for the structure without Z-DON and from PDB ID: 8OXX for the structure with Z-DON. The loops comprising

residues 133-142 and 188-197 in Fab heavy chain B had weak density in both structures and were removed. The structures were submitted to the PDB-REDO web server[44] for further evaluation, and the suggested refinement parameters were used in REFMAC5 to further improve the geometry of the models. The final structures were validated using MolProbity[45] and by careful assessment of the PDB validation report. All figures were prepared with PyMol v3.1.1 (Schrödinger LLC).

## Pull-down of TG3 by anti-TG3 antibodies

Complexes between TG3 and antibodies targeting different TG3 epitopes were generated by incubating 1 μg dispase I-digested TG3 with 2 μg antibody for 30 min in 20 μl PBS. The complexes were then isolated by addition of 10 μl protein G dynabeads (Thermo). After 30 min incubation, the beads were washed several times with PBS supplemented with 0.1% (v/v) Tween-20, using a magnetic separator. The bound proteins were then eluted with 2X SDS sample buffer and analyzed by non-reducing SDS-PAGE.

## Stimulation of human T-cell lines

Gluten-reactive T-cell lines obtained from duodenal biopsies of CeD patients were used for setting up a $^3$H-thymidine proliferation assay essentially as previously described[46]. Briefly, 75,000 antigen presenting cells (HLA-DQ2.5 positive Raji lymphoma cells) were irradiated (90 Gy) and loaded with 10 μg/ml chymotrypsin-digested gluten or medium only before addition of 50,000 T cells and incubation for 48 h at 37 °C. The gluten digest was either added directly or pre-treated with active TG2 or TG3 by incubating 1 mg/ml gluten with 50 μg/ml TG2 or 150 μg/ml cathepsin L-digested TG3 in 100 mM Tris-HCl, pH 7.4, 2 mM $CaCl_2$ for 1.5 h at 37 °C. For direct comparison of TG2 and TG3, 1 mg/ml chymotrypsin-digested gluten was incubated with 50 μg/ml TG2 or 50 μg/ml dispase I-cleaved TG3. After 20, 45 or 90 min, the reaction was stopped by heating for 5 min at 75°C, and the samples were added to 75,000 irradiated (80 Gy) antigen presenting cells (EBV-immortalized B-LCL from an HLA-DR3, HLA-DQ2.5 homozygous donor) to reach a final gluten concentration of 100 μg/ml. The next day, gluten-reactive T-cell lines were added as described above. T-cell proliferation was assessed by pulsing the cultures with 1 μCi $^3$H-thymidine, and measurement of CPM (counts per minute) 16-20 h later.

## Stimulation of BW58 hybridoma T cells

Murine BW58α⁻β⁻ hybridoma T cells expressing human CD4 and gluten-specific TCRs were used to assess gluten peptide deamidation by TG2 and TG3. Activation of DQ2.5-glia-α2-specific (TCR364.1.0.14) and DQ2.5-glia-ω2-specific (TCR737.30) T cells was assessed essentially as previously described[47,48]. Briefly, 50,000 A20 B cells expressing HLA-DQ2.5 (but no transduced BCR) were loaded with different concentrations of synthetic gluten peptides harboring the relevant T-cell epitopes in either their native (Gln) or deamidated (Glu) form before addition of 25,000 TCR-transduced T cells and incubation at 37 °C for 16-20 h. T-cell activation was assessed by measuring release of murine IL-2 into the culture supernatant by ELISA. Native peptides were either added directly or pre-treated with active TG2 or TG3. For comparison of TG2- and TG3-mediated deamidation, 50 μg/ml TG2 or dispase I-digested TG3 was incubated with 200 μM native peptide in 100 mM Tris-HCl, pH 7.4, 2 mM $CaCl_2$ for 1 h at 37 °C before the reaction was stopped by diluting the samples in 5% FBS/RPMI-1640. To test the effect of anti-TG3 antibodies on TG3-mediated deamidation, 40 μg/ml dispase I-digested TG3 was incubated with 200 μM native gluten peptide in the presence or absence of 80 μg/ml anti-TG3 antibody for 25 min at 37 °C, before each sample was diluted with medium and added to the cells in different concentrations. In one set of experiments, the amount of anti-TG3 antibody was varied while the TG3 concentration was kept constant to assess dose-dependent effects of epitope 3-targeting antibodies.

## BCR-transduced A20 cells

BCRs were expressed as transmembrane human IgD molecules by retroviral transduction of murine A20 lymphoma B cells expressing HLA-DQ2.5 as previously described[49,50]. Heavy and light chain sequences of anti-TG3 antibodies were obtained as synthetic DNA (GenScript) and subcloned into the IgD expression vector. To obtain homogenous populations of BCR-expressing cells, transduced A20 cells were stained with anti-human IgD-PerCP/Cy5.5 (1:100, clone IA6-2, BioLegend) in combination with recombinant TG3, and double positive cells were isolated through several rounds of fluorescence-activated cell sorting (FACS), using a FACSMelody instrument (BD). Tetramerized recombinant TGs used for cell staining were generated by incubating $100 \mu g/ml$ biotinylated TG2 and TG3 with $40 \mu g/ml$ streptavidin-APC (Agilent) and streptavidin-PE (Thermo), respectively. After 45 min, free binding sites were blocked by addition of free D-biotin to $20 \mu M$, and the tetramers were added to the cells in PBS supplemented with 2% (v/v) FBS to reach a final concentration of $3 \mu g/ml$ TG2 or TG3. Binding to monomeric TG3 was assessed by incubating the cells with $5 \mu g/ml$ zymogen TG3 in RPMI-1640 on ice followed by washing and incubation with streptavidin-PE in 2% FBS/PBS. Flow cytometry data were analyzed in FlowJo v10 (BD).

## T cell-B cell collaboration assay

To assess BCR-mediated uptake and presentation of gluten peptide, BCR-transduced A20 B cells were incubated with $2.5 \mu g/ml$ TG enzyme (TG2, zymogen TG3 or dispase I-digested TG3) and various concentrations of native gluten peptide at 5 million cells per ml in RPMI-1640 supplemented with 2 mM $CaCl_2$. After 30 min incubation at 37 °C, the cells were washed with RPMI-1640 to remove free peptide. The antigen-pulsed B cells were further incubated 1-2 h in 5% FBS/RPMI-1640 before addition of 25,000 TCR-transduced BW58 T cells to 100,000 A20 B cells. T-cell activation was assessed by measuring IL-2 release after co-culture of the cells as described above.

## Western blotting

To assess TG3-mediated crosslinking of BCRs, BCR-transduced A20 B cells were incubated with $5 \mu g/ml$ zymogen or dispase I-digested TG3 for 30 min at 37 °C in RPMI-1640. The cells were then washed with RPMI-1640 and lysed by re-suspension in $300 \mu l$ 50 mM Tris-HCl, pH 8, 150 mM NaCl supplemented with 1% (w/v) n-Dodecyl β-D-maltoside (Sigma) and cOmplete protease inhibitors (Roche). After incubation for 1 h at 4 °C, the lysate was centrifuged at 15,000 g for 15 min, and $20 \mu l$ of the supernatant was subjected to reducing SDS-PAGE followed by semi-dry transfer onto nitrocellulose. In one set of experiments, denatured lysate was treated with PNGase F (Promega) or O-glycosidase and neuraminidase (New England Biolabs) or a combination of the enzymes for 2 h at 37 °C prior to loading on the gel. After transfer, the membrane was blocked in PBS supplemented with 3% (w/v) dry milk, and the BCRs were detected with rabbit anti-human IgD (1:4000, Dako) followed by goat anti-rabbit Ig-HRP (1:3000, SouthernBiotech) in blocking buffer. For detection of control protein, the membrane was stripped and re-probed with rabbit anti-β-actin (1:1000, clone D6A8, Cell Signaling Technology) followed by goat anti-rabbit Ig-HRP. Signals were obtained by incubation in West-Pico PLUS substrate (Thermo) followed by detection of chemiluminescence on a G:BOX gel doc system (Syngene).

## B-cell uptake of gluten peptide

BCR-transduced A20 B cells were incubated with or without $5 \mu g/ml$ zymogen or dispase I-digested TG3 in RPMI-1640 for 30 min on ice. The cells were then washed and re-suspended in RPMI-1640 supplemented with $5 \mu M$ FITC-labeled native gluten peptide and 2 mM $CaCl_2$ followed by incubation for 30 min at 37 °C. The peptide-loaded cells were washed with 2% FBS/PBS and kept on ice until they were analyzed by flow cytometry, using an LSRFortessa instrument (BD).

## Calcium flux assay

BCR-transduced A20 B cells were loaded with $2 \mu M$ of the cell-permeant $Ca^{2+}$ indicator Fluo-4 (Thermo) in 10% FBS/RPMI-1640 supplemented with 0.02% (w/v) pluronic F-127 (Thermo) for 45 min at 37 °C, before they were washed and re-suspended in 2% FBS/RPMI-1640. The cells were kept on ice but were transferred to a 37 °C water bath 5 min before they were analyzed by flow cytometry on an LSRFortessa instrument (BD). After recording the baseline signal for 30 s, the cells were stimulated by addition of TG3 (zymogen or dispase I-digested) or anti-human IgD F(ab')2 (SouthernBiotech) to a concentration of $5 \mu g/ml$, and recording was continued until reaching 5 min.

## Reporting summary

Further information on research design is available in the Nature Portfolio Reporting Summary linked to this article.

## Data availability

Crystal structures of Fab DH63-A02 bound to TG3 with and without the substrate-mimicking inhibitor Z-DON attached to the active site have been deposited at the Protein Data Bank with PDB IDs 8RMY and 8RMX, respectively. The previously published structures 8OXX and 8OXW were used for comparison. Plasmids and cell lines generated in this study are available from the corresponding authors upon request. Source data are provided with this paper.

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

## Acknowledgements

We thank Marie K. Johannesen and Bjørg Simonsen for technical assistance. We would also like to thank the staff of the ESRF and EMBL Grenoble for assistance and support in using beamline ID23-1 under proposal number MX-2401. We acknowledge MAX IV Laboratory for

time on Beamline BioMAX under Proposal 20220662. Flow cytometry and cell sorting experiments were conducted at the Flow Cytometry Core Facility, Oslo University Hospital. The work was supported by grants from the South-Eastern Norway Regional Health Authority (project 2020027), Stiftelsen KG Jebsen (project SKGJ-MED-017), LEO Foundation (project LF-OC-23-001291) and the University of Oslo World-leading research program on human immunology (WL-IMMU-NOLOGY) to L.M.S.

## Author contributions

Conceptualization, R.I., J.E.H., and L.M.S.; Methodology, R.I., J.E.H., and L.S.H.; Investigation, R.I., J.E.H., S.D., and L.S.H.; Writing – Original Draft, R.I., J.E.H., and L.M.S.; Writing – Review & Editing, all authors; Supervision, L.M.S.; Funding Acquisition, L.M.S.

## Competing interests

The authors declare no competing interests.
