## [Transparent Peer Review file · Nature Communications]

Enzyme-activating B-cell receptors boost antigen presentation to pathogenic T cells in gluten-sensitive autoimmunity

Corresponding Author: Dr Rasmus Iversen

Version 1:

Reviewer comments:

Reviewer #1

(Remarks to the Author)

Iversen and colleagues have addressed appropriately my comments on their manuscript entitled "Enzyme-activating B-cell receptors boost antigen presentation to pathogenic T cells in gluten-sensitive autoimmunity". Although the results are convincing, it should be noted however that most of the results are representative of one out of two experiments and that the data presented have not been analyzed using any statistical methods.

Reviewer #2

(Remarks to the Author)

The authors have revised the manuscript according to the initial comments and I have no further feedback.

Reviewer #3

(Remarks to the Author)

This manuscript is a revised version of a manuscript that I previously reviewed for Nature Immunology. I am of the opinion that the authors have made an adequate response to my previous comments, in particular they have now clarified that this work represents a significant advance over a previous work.

In their response the authors write " For the first time, we both provide evidence that BCR mediated enhanced enzyme activity leads to enhanced antigen presentation, and we provide the molecular explanation for the effect at atomic resolution. Of note, the rheumatoid arthritis study (ref 23) did not include any structural information, nor did it assess T cell-B cell collaboration. "

I agree with this.

Reviewer #4

(Remarks to the Author)

As part of this manuscript the authors report crystal structures of Fab DH63-A02 in complex with cleaved human transglutaminase 3 (TG3), with and without TG3 inhibitor. Binding of inhibitor has been previously shown to induce conformational changes that cause the TG3 C1 and C2 domains to dissociate. Here, in both the inhibitor bound and free structures of the A02 bound TG3, the C1 and C2 domains are also missing. The C1 and C2 domains are presumed to be dislodged by antibody binding and subsequent conformational change. Comparison of these structures with that of an antibody to a different epitope (DH63-B02) bound to non-inhibited TG3 where C1 and C2 are present indicates that DH63-A02 would clash with the C1 domain in its non-inhibited location. The structures need additional refinement to improve their geometry (both structures have a very large number of bond angle, planarity and side chain rotamer outliers as detailed in

the PDB validation reports), and please address other concerns listed below.

Specific comments:

In Supplementary Fig. 2, does the electron density shown come from a 2Fo-Fc map, or an omit 2Fo-Fc map? To show ligand density you should use an SA omit 2Fo-Fc map; please change figure or legend, whichever is required.

Supp. Fig. 3. There are no figures showing locations of the different epitopes recognized by DH63-B02 and DH63-A02. In this figure, please include the Fab B02 instead of removing it, so that it is clear that these two Fabs bind very different regions of the protein.

In Supp. Table 1., it seems odd that the beta angle for 8RMX is ~10 degrees different from that of 8RMY, especially as these crystals were grown using similar conditions and with cross-seeding. Please double check that the beta angle values are correct.

As detailed in the PDB validation report for 8RMX, there are a large number of atoms with zero occupancy. If you really mean to have atoms with zero occupancy, please add a comment in the methods section detailing which atoms have been modeled with zero occupancy and why. Else please give these atoms an occupancy of 1.0 and refine their B values.

Version 2:

Reviewer comments:

Reviewer #4

(Remarks to the Author)

The authors have only uploaded one out of two required validation files. However, the validation report for 8RMX (2.8Å resolution) shows that the structure still has 83(!) bond angle outliers and 127(!) non-rotameric side chains. Just as one example, Thr E15 has a bond angle of 83.3 deg for CA-CB-OG1. This angle should be 109 deg, we know what this angle should be, it is a given and the refinement software should be weighting things properly so that this angle comes out very close to 109. These sorts of errors should not be a problem if the weight between Fo and Fc is chosen correctly, it has nothing to do with the flexibility of the molecule. If you can't get Refmac to put out a model with better geometry you should contact the Refmac developers to make sure there is not an issue with the Refmac version you are using. The numbers of outliers in your deposition 8OXX (2.5Å) that also is a Fab-transglutminase3 complex at a similar resolution, are acceptably low, so I don't think there is an issue with these particular proteins.

So please check how you are running Refmac, the Fo-Fc weight needs to be adjusted to tighten up the geometry, else there is a problem with the version of the software you are using and you need to check with the developers.

If you resubmit this again include both validation reports.

Version 3:

Reviewer comments:

Reviewer #4

(Remarks to the Author)

Thanks to the authors for the re-refinement. The structure parameters look much better now.

Reviewers' comments:

Reviewer #1 (Remarks to the Author):

In this manuscript entitled “Enzyme-activating B-cell receptors boost antigen presentation to pathogenic T cells in gluten-sensitive autoimmunity”, Iversen and colleagues describe a mechanism by which autoantibodies against the enzyme transglutaminase 3 could contribute to the pathogenesis of dermatitis herpetiformis by enhancing TG3 enzymatic activity that modify and cross-link gluten peptides enhancing their presentation to T cells.

They first demonstrate that gluten peptides deamidated by TG3, particularly ω 2-gliadin, can be taken up by B cells, likely through a mechanism involving the uptake by the BCR of TG3-gluten complexes, to promote T cell activation. This is achieved by showing that i) stimulation of human T cell lines obtained from celiac disease patients with TG3-treated gluten leads to a higher level of activation as compared to stimulation with untreated gluten; ii) stimulation of mouse T cells specific for DQ2.5-glia- ω 2 by HLA-DQ2.5-expressing A20 B cells in the presence of TG3-treated gluten peptides promotes T cell activation; and iii) stimulation of mouse T cells specific for DQ2.5-glia- ω 2 by A20 B cells expressing an anti-TG3 IgD BCR in addition to HLA-DQ2.5 in the presence of gluten peptides and TG3 also leads to T cell activation. This mechanism of presentation of enzymatically modified gluten peptides resembles the mechanism described by the authors for TG2. Next, they show that binding of anti-TG3 antibodies to one group of conformational epitopes within TG3 (epitope group 3) leads to the dissociation of the C1C2 domains of TG3 from the catalytic domain triggering the catalytically active enzyme conformation. Analysis of the structure of an epitope group 3 Fab bound to cleaved TG3 confirmed that the C-terminal domains are lacking and that the anti-TG3 antibody binds to the catalytic core β -sheet via electrostatic interactions with residues present in heavy and light chains CDRs suggesting that antibody binding promotes a conformational change that uncovers the active site and therefore enhances the catalytic activity of the enzyme. Finally, they show that BCR targeting the epitope 3 binds to TG3 promoting B cell activation as shown by measuring release of intracellular calcium and enhances gluten presentation to T cells when BCR-transduced A20 cells representing epitope 3 and T cells were co-cultured in the presence of gluten and cleaved TG3.

This a well written manuscript with straightforward experiments and convincing results that uncover a new role for TG3 and anti-TG3 antibodies that have important implications for the pathogenesis of dermatitis herpetiformis. Particularly exciting is the finding that antibodies can contribute to disease development by affecting the activity of the enzyme TG3 and the ensuing T cell-B cell crosstalk.

Response: We thank the reviewer for the positive comments about our study.

Specific comments:

- In Figure 1, it would be important to determine whether deamidation of gluten peptides is efficiently achieved with TG3 and whether it as efficient as with TG2.

Response: By using whole gluten and stimulation of oligoclonal T-cell lines specific to deamidated epitopes, we demonstrate that both TG2 and TG3 can generate deamidated peptides that are recognized by gluten-reactive T cells (Fig. 1a). To compare the efficiency of TG2- and TG3-mediated deamidation, we have set up new experiments, where gluten was incubated with either TG2 or TG3 for various amounts of time, before it was used to stimulate T-cell lines. These data are shown in Supplementary Fig. 1 of the revised manuscript. The results show that TG2 and TG3 deamidate relevant gluten epitopes with comparable efficiency.

- In Figure 1a, it is indicated that “bar heights indicate means of culture triplicates”. Is it culture triplicate from one experiment only or from several experiments? The figure legend is misleading. If only one experiment is presented in the main figure, experiments with similar results should be shown as supplemental data.

Response: The figure shows data from one experiment. We have now clarified this in the figure legend. One more T-cell line has been included in Fig. 1a, and we have repeated the experiment using additional T-cell lines of which one was also used in the first experiment. These data are shown in Supplementary Fig. 1 (see also reply to the point above). Importantly, all tested T-cell lines showed a response to both TG2- and TG3-treated gluten, but not to untreated gluten.

- In Figure 2, more context should be given regarding the isolation of the ten antibodies tested and the epitopes they target.

Response: The isolation of the antibodies has been reported previously (PMID: 37424036), and the study is referenced in the Introduction. We have now also included relevant background information in the Results section, where Fig. 2 is described.

- In Figure 4a, the authors should explain the difference in TG3 binding between DH63-A02 and the other BCRs.

Response: The lower level of TG3 binding likely results from a combination of lower BCR expression (as shown in Fig. 4a) and lower affinity for TG3. We have clarified this in the text.

- Given that the formation of complexes between TG3 and gluten peptides enhances gluten presentation and therefore T cell activation, the authors should expand the interpretation of the results and discuss the role of these activated pathogenic T cells in the context of dermatitis herpetiformis. In addition, they should comment on the location of these interactions between TG3, gluten, and B cells.

Response: We agree with the reviewer that the role of pathogenic T cells is a very relevant point. It is well established that gluten-reactive CD4+ T cells are important for establishing the gut lesion in celiac disease. It is possible that they also play a role in formation of skin blisters in dermatitis herpetiformis and that local T cell-B cell interactions are involved. This possibility, however, has so far not been studied. We have included these considerations in the Discussion.

Reviewer #2 (Remarks to the Author):

The paper deals with DH, a skin disease that occurs in a subset of patients with celiac disease. The authors show that TG3-specific B cells, typically present in patients with DH, can present deamidated gluten peptides through BCR-mediated uptake and subsequent processing of TG3-gluten complexes. This has been tested with BCR-transduced A20 mouse lymphoma cells, not with B cells isolated from patients. Moreover, this is a mechanism principally identical to what has previously been demonstrated for TG2-specific B cells in celiac disease patients. So there is limited novelty here. The authors also show that anti-TG3 antibodies can enhance gluten deamidation and that is due to an antibody induced conformational change in the TG3 enzyme as shown in functional experiments and supported by co-crystal structures between the antibody and the enzyme. Although this is a very

nice demonstration of how such antibody mediated enzyme augmentation can take place, this concept in autoimmunity is not new as this was already shown to play a role in rheumatoid arthritis in 2013 (ref 23 in the manuscript), as acknowledged by the authors. So I am afraid that there is limited novelty here as well.

Response: As the reviewer correctly points out, we have used BCR-transduced B-cell lines rather than B cells isolated from patients. Importantly, though, the BCR sequences were obtained from plasma cells of DH patients and the recombinant BCRs are thus disease relevant. Ideally, we would have used B cells isolated directly from patients to set up T cell-B cell collaboration assays. However, antigen-specific B cells cannot easily be obtained from patient samples in sufficient numbers to set up such assays, and we therefore used transduced cell lines. We respectfully disagree with the reviewer that the concept is not new in autoimmunity. For the first time, we both provide evidence that BCR-mediated enhanced enzyme activity leads to enhanced antigen presentation, and we provide the molecular explanation for the effect at atomic resolution. Of note, the rheumatoid arthritis study (ref 23) did not include any structural information, nor did it assess T cell-B cell collaboration.

Reviewer #3 (Remarks to the Author):

The manuscript entitled “Enzyme-activating B-cell receptors boost antigen presentation to pathogenic T cells in gluten-sensitive autoimmunity” submitted to Nature Immunology by Iversen and colleagues addresses the question whether TG3-autoantibody specific B cells characteristic for the skin manifestation of celiac disease, dermatitis herpetiformis (DH) can be activated by gluten-specific CD4+ T cells. The authors also demonstrate that the TG3-autoantibodies enhance the catalytic activity of TG3, and this effect translates into increased gluten presentation to T cells when such antibodies are expressed as BCRs. The laboratory work has been carefully conducted with proper controls allowing the conclusion to be made.

The findings represent a significant advance in the field the field of DH research and the TG3-specific B cells seem to act largely similarly than TG2-specific B cells in celiac disease which has already earlier been established. It is of note that DH affects 1/8 of celiac disease patients and thus 0.25% of the general population. Therefore the DIRECT generalizability remains of the findings remain limited. However, the authors postulate that similar mechanisms might also apply in other autoimmune diseases, for instance rheumatoid arthritis.

Response: We appreciate the positive comments by the reviewer. We agree that the condition we focus on, DH, is rare. Still, the concept that BCR binding affects the structure and function of a self-antigen in a way that enhances T cell-B cell crosstalk is novel, and we believe that it could have implications for other autoimmune diseases.

Below are points that the manuscript raises.

The authors use T cell activation assay to assess TG3-mediated deamidation of gliadin peptides (Figure 1 and page 5). As the used T cell lines require deamidation, the selected system is suitable. However, the current results do not reveal the site specificity of the deamidation of the alpha2 and omega 2 gliadin peptides which would be interesting. Although a previous article by Stamnaes et al. (doi: 10.1007/s00726-010-0554-y) touches upon this issue in terms of the alpha2 gliadin, the precise

amino acid targeting of TG3-mediated deamidation of both the alpha2 and gamma2 peptides would add important information to the article.

Response: The T-cell epitopes have been accurately defined, so it is known exactly which residues require deamidation. We acknowledge that there could be other glutamine residues that will be targeted differentially by TG2 and TG3. This possibility is described in the Discussion. In the relevant assay, we use highly specific T-cell clones, meaning that the readout is set on those residues that are required for T-cell activation. We have clarified this aspect in the text.

In figure 1d-e, the authors use DH63-B02 for testing if TG3-specific B cells can present gluten peptides to gluten-specific T cells. The DH63-B02 antibody targets epitope 2 in TG3, while according to their previous work (Das et al. doi: 10.1002/adv.202300401) and in the introduction (page 4, lines 75-76) the authors raised the idea that epitope 3 antibodies are the ones dominating the TG3-antibodies in DH. Thus, why was epitope 2 antibody selected in this context? It would be important to conduct the same experiments with also epitope 3 (and also) epitope 1 antibodies.

Response: We started out using DH63-B02, since we had previously characterized the binding between this antibody and TG3, and we knew the location of the epitope (PMID: 37798283). We have clarified this rationale in the text. Importantly, the same experiments were conducted with BCRs targeting all three epitopes to compare their effect on antigen presentation. These data are reported in Fig. 4h.

The rationale that TG3-specific B cells would present gliadin peptides to T cells, would require the presence of TG3-gliadin peptide complexes. This is indirectly addressed in the article by demonstrating of gliadin peptide uptake of B cells loaded with TG3. Stamnaes et al. (doi: 10.1007/s00726-010-0554-y) have reported that TG3 and gliadin only form thioester-linked but not iso-peptide complexes. How does this fit into the scheme of the current manuscript? This issue needs to be discussed and preferably also addressed by experiments.

Response: We believe that the thioester-linked complex is the most relevant gluten-TG complex for both TG2 and TG3. As the reviewer points out, it was previously demonstrated that recombinant TG3 (unlike TG2) does not form isopeptide-linked complexes with gluten. We have also previously demonstrated that endogenous TG2 in epithelial cell lysates does not form such complexes, although the enzyme can drive T cell-B cell collaboration under the same conditions (PMID: 32302613). These findings point to the enzyme-substrate intermediate as the entity that allows TG-specific B cells to interact with gluten-specific T cells. We have now included these considerations in the Discussion.

To study the effects of TG3-antibodies to TG3-mediated deamidation, the authors chose all 4 previously identified epitope 1 and 2 antibodies but yet only included 2/4 of the epitope 3 antibodies. Although the 2 untested epitope 3 antibodies were derived from patient DH44 from whom also the tested DH44-A10 was, the 2 other epitope 3 antibodies should also be tested for TG3-mediated deamidation particularly because the authors speculate that epitope 3 antibodies are the dominating ones in DH.

Response: The two epitope 3 antibodies that were not tested belong to the same clonotype as DH44-A10. Thus, we only had two epitope 3 clonotypes available, both of which were tested as soluble antibodies and BCRs. We have provided more details about the anti-TG3 antibodies in the Results section.

It is starting to be a general practice in many journals that in figures representing gel electrophoresis

the entire gel (instead of a selected area) is shown – I would therefore urge to authors to adopt that practise.

Response: The uncropped gel pictures are included with the Source Data accompanying the paper.

The literature on TG3-plasma cells and T cells in DH is not yet very immense, so I would encourage the authors to include as reference the finding of TG3-specific intestinal plasma cells (doi: 10.2340/00015555-2849.) and the first article on peripheral gluten-specific cells (doi: 10.1016/j.jid.2019.12.038)

Response: We appreciate the suggestion. The references are now included in the manuscript.

Nature Communications manuscript NCOMMS-24-60943A

Point-by-point response to comments of Reviewer #4:

As part of this manuscript the authors report crystal structures of Fab DH63-A02 in complex with cleaved human transglutaminase 3 (TG3), with and without TG3 inhibitor. Binding of inhibitor has been previously shown to induce conformational changes that cause the TG3 C1 and C2 domains to dissociate. Here, in both the inhibitor bound and free structures of the A02 bound TG3, the C1 and C2 domains are also missing. The C1 and C2 domains are presumed to be dislodged by antibody binding and subsequent conformational change. Comparison of these structures with that of an antibody to a different epitope (DH63-B02) bound to non-inhibited TG3 where C1 and C2 are present indicates that DH63-A02 would clash with the C1 domain in its non-inhibited location. The structures need additional refinement to improve their geometry (both structures have a very large number of bond angle, planarity and side chain rotamer outliers as detailed in the PDB validation reports), and please address other concerns listed below.

Response: We thank the reviewer for this detailed feedback on the structural data. We agree that the structures have a significant number of geometric outliers, but the structures have been through over 200 iterations of trying to improve this, using Molprobit to assess each residue and its H-bonding interactions. Parts of the structures have poor density. In these regions, the improvements on geometry made during manual assessment were disrupted upon refinement. We have removed the worst part of the loops, but do not want to remove larger parts of the structures as this can cause scientists with limited structural knowledge to draw incorrect conclusions. An illustration of this problematic issue can be seen in the otherwise very nice article by Hang et al. 2005 (PMID: 15849356, Figure 7), where the authors describe an accessible “hairpin loop” in the N-terminal domain of the transglutaminase 2 structure PDB ID: 1KV3. This in fact is an artefact of the removal of three loops of TG2 that probably displayed weak electron density (residues 1-14, 44-55 and 121-132). There are of course different opinions on the removal of poor-density residues in the crystallographic community, but with the increasing use of structural data for drug design etc., we are of the opinion that it is productive to leave such regions in as much as possible and accept the poorer validation.

Specific comment 1:

In Supplementary Fig. 2, does the electron density shown come from a 2Fo-Fc map, or an omit 2Fo-Fc map? To show ligand density you should use an SA omit 2Fo-Fc map; please

change figure or legend, whichever is required.

Response: As requested, we have generated an omit map (from the data right after molecular replacement, before the model ever saw the ligand, hence unbiased) and have updated the figure to include this. The figure legend is also updated with more information about the maps displayed.

Specific comment 2:

Supp. Fig. 3. There are no figures showing locations of the different epitopes recognized by DH63-B02 and DH63-A02. In this figure, please include the Fab B02 instead of removing it, so that it is clear that these two Fabs bind very different regions of the protein.

Response: We are grateful for this suggestion. Supplementary Fig. 3 has now been updated with additional panels showing the binding sites of Fab DH63-B02 and Fab DH63-A02 as well as an overlay of the two Fab-bound structures.

Specific comment 3:

In Supp. Table 1., it seems odd that the beta angle for 8RMX is ~10 degrees different from that of 8RMY, especially as these crystals were grown using similar conditions and with cross-seeding. Please double check that the beta angle values are correct.

Response: We appreciate the rigorous investigation of our data. It is correct that the beta angles of the unit cells differ. By comparing the two structures and contents of the unit cell, we observe that the two copies of the complex in the asymmetric unit are marginally skewed relative to each other. This might have caused the slight shift in angle for the optimal solutions picked by the processing software.

Specific comment 4:

As detailed in the PDB validation report for 8RMX, there are a large number of atoms with zero occupancy. If you really mean to have atoms with zero occupancy, please add a comment in the methods section detailing which atoms have been modeled with zero occupancy and why. Else please give these atoms an occupancy of 1.0 and refine their B values.

Response: We thank the reviewer for noticing this. The atoms with zero occupancy were riding hydrogens added by reftmac to facilitate geometry improvements. They should not be

part of the final deposition, although they do not change the model. We apologize for this glitch. We have now uploaded the same model without the hydrogen atoms to PDB and included the updated validation report with our submission. The validation statistics have also slightly improved as a result of this, so we are grateful for this correction.

REVIEWER COMMENTS

Reviewer #4 (Remarks to the Author):

The authors have only uploaded one out of two required validation files. However, the validation report for 8RMX (2.8Å resolution) shows that the structure still has 83(!) bond angle outliers and 127(!) non-rotameric side chains. Just as one example, Thr E15 has a bond angle of 83.3 deg for CA-CB-OG1. This angle should be 109 deg, we know what this angle should be, it is a given and the refinement software should be weighting things properly so that this angle comes out very close to 109. These sorts of errors should not be a problem if the weight between F_o and F_c is chosen correctly, it has nothing to do with the flexibility of the molecule. If you can't get Refmac to put out a model with better geometry you should contact the Refmac developers to make sure there is not an issue with the Refmac version you are using. The numbers of outliers in your deposition 8OXX (2.5Å) that also is a Fab-transglutminase3 complex at a similar resolution, are acceptably low, so I don't think there is an issue with these particular proteins.

So please check how you are running Refmac, the F_o - F_c weight needs to be adjusted to tighten up the geometry, else there is a problem with the version of the software you are using and you need to check with the developers.

If you resubmit this again include both validation reports.

Response: We thank the reviewer again for a thorough evaluation of our data. We have worked on improving the quality of both structures. It turned out to be helpful to submit the structures to the PDB-redo web server and use the resulting refinement parameters in CCP4 Refmac5 together with stricter geometry weights. This additional exercise is now mentioned in the Methods. We also tried the refinement program Servalcat and the low-resolution refinement pipeline LORESTR, but neither of these programs did significantly improve the geometry compared to using the PDB-redo parameters together with manual adjustments in Coot. Importantly, now the scores for Rfree, Clash, Ramachandran, sidechain and RSRZ outliers are all better than the average when compared to X-ray structures of similar resolution in PDB. The validation reports for both structures are included with this resubmission.